# Role of Non-Binding T63 Alteration in IL-18 Binding

**DOI:** 10.3390/ijms252312992

**Published:** 2024-12-03

**Authors:** Chariya Peeyatu, Napat Prompat, Supayang Piyawan Voravuthikunchai, Niran Roongsawang, Surasak Sangkhathat, Pasarat Khongkow, Jirakrit Saetang, Varomyalin Tipmanee

**Affiliations:** 1Department of Biomedical Sciences and Biomedical Engineering, Faculty of Medicine, Prince of Songkla University, Songkhla 90110, Thailand; 6210320002@email.psu.ac.th (C.P.); napatprompat@gmail.com (N.P.); surasak.sa@psu.ac.th (S.S.); pasarat.k@psu.ac.th (P.K.); 2Center of Antimicrobial Biomaterial Innovation-Southeast Asia and Natural Product Research Center of Excellent, Faculty of Science, Prince of Songkla University, Songkhla 90110, Thailand; supayang.v@psu.ac.th; 3Microbial Cell Factory Research Team, National Center for Genetic Engineering and Biotechnology, National Science and Technology Development Agency, Pathum Thani 12120, Thailand; niran.roo@biotec.or.th; 4Department of Surgery, Faculty of Medicine, Prince of Songkla University, Songkhla 90110, Thailand; 5Translational Medicine Research Center, Faculty of Medicine, Prince of Songkla University, Songkhla 90110, Thailand; 6Institute of Biomedical Engineering, Faculty of Medicine, Prince of Songkla University, Songkhla 90110, Thailand; 7EZ-Mol-Design Laboratory, Faculty of Medicine, Prince of Songkla University, Songkhla 90110, Thailand; 8International Center of Excellence in Seafood Science and Innovation, Faculty of Agro-Industry, Prince of Songkla University, Songkhla 90112, Thailand

**Keywords:** interleukin-18, interleukin-18 receptor, molecular dynamics simulation, point mutagenesis, protein conformation, protein dynamics

## Abstract

Engineered interleukin-18 (IL-18) has attracted interest as a cytokine-based treatment. However, knowledge-based mutagenesis of IL-18 has been reported for only a few regions of the protein structures, including binding sites I and II. When coupled with the binding region mutant (E6K), the non-binding residue of IL-18, Thr63 (T63), has been shown to increase the flexibility of the binding loop. Nevertheless, the function of Thr63 in conformational regulation is still unknown. Using homology modeling, molecular dynamics simulation, and structural analysis, we investigated the effects of Thr63 alteration coupling with E6K on conformational change pattern, binding loop flexibility, and the hydrogen bond network. The results indicate that the 63rd residue was significantly associated with hydrogen-bond relaxation at the core β-barrel binding sites I and II Glu85-Ile100 loop. This result provided conformational and flexible effects to binding sites I and III by switching their binding loops and stabilizing the 63rd residue cavity. These findings may pave the way for the conceptualization of a new design for IL-18 proteins by modifying non-binding residues for structure-based drug development.

## 1. Introduction

Interleukin-18 (IL-18) is a well-known immunoregulator because it stimulates the production of interferon-γ (IFN-γ) by Th1 cells, natural killer (NK) cells, natural killer T (NKT) cells, B cells, dendritic cells (DCs), and macrophages [1]. IL-18 also acts as an inducer of anticancer activity via cytokine or adoptive cell therapies [2]. Prior to clinical trials of novel therapies, however, treatment efficacy and toxicity must be modified [3]. As a result, cytokine engineering appears to be a promising method for improving immunotherapies. In preclinical models, an engineered decoy-resistant IL-18 (DR-18) capable of evading IL-18 binding protein (IL-18BP) demonstrated potent antitumor activity [4]. T cells that have been redirected for antigen-unrestricted cytokine-initiated killing (TRUCK) also deliver IL-18 to tumor tissue. Thus, TRUCKs may hold the key to increasing the efficiency of chimeric antigen receptor (CAR) T cells through a variety of mechanisms [5,6,7,8].

IL-18 plays important roles in immune homeostasis [9], and its functions have been identified in pro-inflammatory cytokines [10,11,12]. Site-directed mutagenesis could be used to identify the critical amino acid residue that influences IL-18 function. The E6 and K53 residues of IL-18 are crucial for IL-18R and IL-18P binding [13,14]. The E6K mutation improves NK cell activation protein function [14,15], while changing Lys53 to glutamic acid (K53E) reduces cytokine activity [16]. In contrast, some mutated IL-18s, D17N and M33Q, altered the stabilization of site I, resulting in a decrease in IFN-γ [15,16]. The double mutant E6K/T63A reported a 16-fold increase in IL-18 activity, and the crucial role of binding loop flexibility due to the presence of non-binding T63 was suggested [16,17].

The coupling effect of binding and non-binding residues was observed in IL-18, IL-2, and IL-1β [18,19,20]. The computationally designed Neo-2/15 or IL-2/IL-15 induced superior IL-2 therapeutic activity [21]. In addition, the simulation of folding and kinetics of F42W/W120F in IL-1β revealed that structural water interaction and multiple folding routes for IL-1β altered signaling activity. Therefore, dynamics networks of IL-2 and IL-1β that were driven by their distal regions investigated the mysteries of its native and bound states. Previous studies have used molecular dynamics (MD) simulation to investigate the structure of IL-18 [16,22,23]. The stability of the IL-18s, IL-18/IL-18R, and IL-18/IL-18BP was evaluated so that they could be used in structure-based molecular drug design [22]. Prior studies found that E6K/T63A IL-18 structure influenced both structural flexibility and interaction sites [16]. However, how the 63rd residue affects protein structure is unknown.

We simulated engineered IL-18s with the E6K/T63X to better understand the structural basis of the 63rd residue’s interaction with E6K. The secondary structure prediction and sequence alignment of human IL-18 and other mammalian IL-18s were investigated. The initial and dynamic structures were generated using homology modeling and MD simulation. Furthermore, the overall structure pattern and traditional hydrogen-bond relaxation revealed intramolecular behaviors. As a result, the effect of modifying non-binding residues on the inner core structure of IL-18 binding sites was investigated.

## 2. Results

### 2.1. Structural Analysis of E6K/T63X by MD Simulation

#### 2.1.1. Sequence Alignment and Secondary Structure Prediction

The full IL-18 protein sequences were used to compare the similarity of E6 and T63 across species. Protein sequences were obtained from the Uniprot database for humans (Q14116), rats (P97636), cattle (Q9TU73), horses (Q9XSQ7), and mice (P70380) (Figure 1).

The E6 is present in humans, cattle, and horses, while rats and mice sequences contain histidine at this location. Despite being in the coil of binding site II (K4-E6), E6 was not a binding residue. T63 is conserved in all selected organisms. The identity percentage of full-length IL-18 sequences in rats, bovines, horses, and mice was 62.1%, 77.7%, 79.3%, and 62.3%. Appendix A shows the results of the protein sequence alignment analysis.

#### 2.1.2. MD Simulation of E6K/T63X and Distance Pattern Study

Although the static IL-18 structures were computationally generated using homology modeling [24], the dynamics condition could provide a more atomistic understanding of engineered IL-18s. All MD simulation systems in this study were stable, as measured by root mean square deviation (RMSD) (Appendix A). Nonetheless, E6K/T63K cannot be simulated by MD simulation because the system cannot be energy-minimized due to the high van der Waals contact between the long lysine sidechain and its surrounding amino acid (or AA) residues. As a result, this mutant was not included in the MD simulation.

We investigated the overall structural change using the center of geometry study. The distance pattern between the 63rd and other residues was used to represent the regions of conformational change (Figure 2). For E6K/T63A, the large different region was located in the large loop of binding site I (Glu31-Thr45), while binding site III loops (Gly108-Lys112 and Asp142-Met150) exhibited minor structural changes (Figure 2). Binding site II region (Lys53-Met60) remained similar across all mutant structures.

The distance between the large binding site I loop in E6K/T63A increased by approximately 5 Å compared to the wild type. E6K/T63A binding site III loops followed the same distant pattern as E6K. The single-mutation variants revealed that E6K had a significant impact on binding sites, whereas T63A retained the wild type structure. However, E6A only moderately promoted this loop, leaving binding site III loops alone. E6K influenced the structural modification of E6K/T63A by altering binding site I and III regions.

The majority of the engineered E6K/T63X IL-18 followed the same pattern as E6K/T63A, including E6K/T63I, E6K/T63L, E6K/T63M, E6K/T63N, E6K/T63Q, E6K/T63W, and E6K/T63D. Nonetheless, E6K-related conformational changes at binding site III were not observed in some mutants, including E6K/T63F, E6K/T63Y, E6K/T63P, E6K/T63H, and E6K/T63R. Furthermore, mutations (E6K/T63V and E6K/T63E) canceled the E6K-related conformational change at site I, resulting in the T63A pattern.

In the case of E6K/T63S, E6K’s effects completely altered only binding site III, leaving binding site I pattern unchanged. The effect at binding site I was almost completely eliminated in E6K/T63C and E6K/T63G. These findings suggested that a combined mutation of E6K at the binding region and a Thr63 alteration at the non-binding region can rearrange the IL-18 structure at the large loop of binding site I and the two loops of binding site III. The 63rd residue substitution in these mutants could have a significant impact on the structural effects of E6K.

#### 2.1.3. Binding Loop Flexibility

The root-mean square fluctuation (RMSF) was used to determine the binding loop flexibility. The results showed that the coupled mutation affected the flexibilities at binding sites I and III. The large binding site I (Glu31-Thr45) of E6K/T63A exhibited higher flexibility than the wild type, whereas the small binding site I (Glu130-Phe134) loop was lower than the wild type (Figure 3). Similarly, binding site III flexibilities of E6K/T63A showed that the RMSF values of the Gly108-Lys112 loop increased while the flexibility of the Asp142-Met150 loop decreased to lower than the wild type.

This binding loop flexibility result could be related to the structural effects of E6K and T63A on the coupled mutant. T63A, on the other hand, had the ability to modify binding site I and III flexibilities without undergoing a conformational change. Furthermore, individual single mutations of E6K and T63A promoted flexibility at binding sites I and III, resulting in improved E6K/T63A flexibility. In the case of E6A, despite switching the loop structure, the large binding site I flexibility was unable to relax. As a result, E6K mutation was used to evolve binding site I flexibilities of the engineered IL-18s. These findings could support the synergistic effects of E6K-T63A, which implemented structural changes and flexibility [16].

The effects of E6K on binding site I flexibility were removed in mutants E6K/T63V, E6K/T63Y, E6K/T63C, E6K/T63H, and E6K/T63E. E6K reduced the flexibility of the majority of the engineered IL-18s (Figure 3). These findings were noteworthy because the distant pattern effects of E6K on structural change and flexibility at binding site I were independent. Interestingly, the flexibility of this site was higher in E6K/T63F and E6K/T63D than in E6K. The high RMSF values at the large binding site I (Glu31-Thr45) loop were found to be a possible feature of the coupled mutant IL-18s for enhancing structural flexibility. In terms of binding site II flexibility, most engineered IL-18s had lower RMSF values than the wild type (Figure 3). E6K/T63I, E6K/T63N, and E6K/T63R, on the other hand, conserved the region’s flexibility in the same way that the wild type did. Furthermore, each mutant’s binding site III loops exhibited a distinct flexibility (Figure 3). The flexibility of this site’s Gly108-Lys112 loop was increased in various mutants compared to the wild type. Nonetheless, this loop flexibility was conserved in E6K/T63V, E6K/T63I, E6K/T63L, E6K/T63M, E6K/T63Q, and E6K/T63D.

This region was in contact with IL-18Rα and formed a helix–loop structure [25]. As a result, the side chain flexibility may represent the receptor’s loop ability to penetrate its binding pocket. Although this region (Lys53-Met60) retained its conformation from the distance pattern study, the flexibility was affected individually by each of its side chains. To stabilize binding site II structure, two β-sheets (Thr45-Tyr52 and Ala61-Cys68) would form a strand (Figure 1). These sheet structures also solidified in terms of conformational change and flexibility. Rather than loop switching, the interaction between binding site II and the receptor was initiated from the side chains of binding residues. Then, another loop at this site (Asp142-Met150) showed great adaptability. With the exception of E6K/T63M, the majority of the mutants had decreased loop flexibility. Furthermore, a few mutants, including E6K/T63L, E6K/T63G, and E6K/T63D, had low RMSF values for this loop, which was same as the β-sheet regions. This loop became more rigid in the E6K/T63K mutant as a result of the conformational change occurring in this region.

### 2.2. Atomistic Feature of the 63rd Residue and Neighbor Residues Through H-Bond Analysis

#### 2.2.1. Conventional H-Bond Analysis

H-bond occupancy was used to assess the significance and occurrence percentage of the bond during the simulation. H-bond formation and deformation can be a useful parameter for tracking secondary structure changes in each region during the dynamics condition [26]. The wild type H-bonds with higher than 80% occupancy were considered as the conventional H-bonds of IL-18. Then, these bonds in the engineered IL-18s were examined to represent the bond relaxation (Figure 4). The results illustrated that the conventional H-bonds (dash lines) were located at the non-binding region, binding site I, binding site II, and the Glu85-Ile99 loop (Figure 4A–E). While there was no conventional H-bond at binding site III, the bonds at the core structure were placed near this binding site (Figure 4B). The percentage of H-bond occupancy was ranked in the heatmap (Figure 4F).

Binding site I, Asp35@O-Arg39@HH22, was established in the middle of the α-helix with 87.8% occupancy in the wild type, but it was nearly broken in the engineered IL-18s (Figure 4C,F). This result supported the possibility of a helix–coil transition occurring in this region. Met51@O-Leu5@H connected two sheets from this site with occupancies greater than 50% for binding site II (Figure 4D,F). Furthermore, the Glu85-Ile100 loop’s Pro89@O-Tyr52@HH appeared to support Met51@O-Leu5@H for binding site II stabilization. Except for E6K/T63F, which reduced the Pro89@O-Tyr52@HH occupancy dramatically to 22.7%, these two H-bonds preserved the conventional bonds in many mutants. These findings also supported the distance pattern and flexibility of binding site II, indicating that the structure was preserved by the two sheets and traditional H-bond stabilization.

The Glu85-Ile100 loop, which acted as a lid to cover the 63rd residue at the IL-18 surface, was another region of conventional H-bonds (Figure 4E). Met86@O-Thr73@HG1, Pro89@O-Tyr52@HH, Thr95@O-Ser118@HG, and Asp98@OD2-Ser75@HG were all involved in the H-bond networks of this region. These bonds played an important role in connecting the lid’s surface residues (Met86, Pro89, Thr95, and Asp98) and the core structure’s non-binding residues (Thr73, Tyr52, Ser118, and Ser75).

Most IL-18 mutants conserved more than 68% of the Met86- and Pro89-associated H-bonds, whereas E6K/T63N, E6K/T63Q, E6K/T63F, E6K/T63Y, E6K/T63H, and E6K/T63R disrupted these bond formations and decreased occupancies to less than 40% (Figure 4F). Thr95@O-Ser118@HG and Asp98@OD2-Ser75@HG occupancies, on the other hand, were lower than 33% in many mutants, with an exception of E6K/T63A, which exhibited occupancies of 49.0% and 53.9%, respectively. These findings suggested that the large side chain of the 63rd residue change could disrupt the stabilization of the IL-18 structure. IL-18’s conventional H-bonds were adaptable in terms of their core structure and binding regions.

#### 2.2.2. The 63rd Residue-Associated H-Bonds and Effects of the 63rd Residue Cavity Transformation

Figure 5 shows the structural insight and direct H-bonds of the 63rd residue and neighbors. Thr63 at the non-binding region was surrounded by hydrophobic residues (Ile48, Ile49, Tyr52, Val62, Ile64, Pro88, Ile92, Ile99, and Ile100) and hydrophilic residues (Ser50, Ser65, and Thr73) (Figure 5A). This region was hydrophobic with a wild type hydrophobicity index of 9.28. The wild type backbone atoms of Thr63 and Ser50 bonded between two β-sheets, Ser50@O-Thr63@H (74.2% occupancy) and Thr63@O-Ser50@H (63.6% occupancy) (Figure 5B). These bonds had fluctuating occupancies from the Thr63 alteration in many mutants, which may affect sheet stabilization. These stabilized bond occupancies were reduced to less than 50% in some large-residue substitutions, including E6K/T63Q, E6K/T63F, E6K/T63Y, E6K/T63P, and E6K/T63D, suggesting that the backbone of the 63rd residue-involved H-bonds was able to loosen the core structure through local H-bond relaxation.

In addition to the Thr63 backbone-related H-bond, the hydroxyl (-OH) group of the Thr63 side chain was involved in the strongest H-bond (Ile99@O-Thr63@HG1) of the wild type for 94.3% occupancy; however, the modification of this residue adapted its side chain with other neighbor residues instead of Ile99, including Thr73 and Tyr52 (Figure 5B). Ile99@O, Thr73@OG1, and Tyr52@OH bent their side chains to surround the 63rd residue and formed H-bonds (Figure 5B). Many mutants used their -OH or amine (-NH2) or amide (-CONH2) or carboxyl (-COOH) group of the side chain as an acceptor atom for the H-bond formation with the Y52/T73/I99 ring, including E6K/T63S, E6K/T63N, E6K/T63Q, E6K/T63W, E6K/T63H, E6K/T63R, E6K/T63D, and E6K/T63E (Figure 5B). Due to occupancies less than 80%, these bonds were unable to function as conventional wild type and E6K bonds. Furthermore, mutants lacking the acceptor atom of the 63rd residue side chain lost all of their H-bonds. These findings indicated that the 63rd residue side chain could bend the direction for assembling H-bonds with neighboring residues.

## 3. Discussion

### 3.1. Binding Site Analysis of Wild Type and Engineered IL-18 via MD Trajectories

Nowadays, engineered IL-18 is a promising biologic that could enhance innate and adaptive immunities and avoid the immune barrier for cancer immunotherapy [3,4]. Several studies have showcased the structure of IL-18 and its specific binding interactions with IL-18R [12,25,27], IL-18 inhibitor [28,29], and IL-18BP [30] for structure-based drug development. However, engineered IL-18 design remains uncertain and there are few data pertaining to conformational regulation. Previously, binding residue substitution of IL-18 was experimentally demonstrated to enhance binding affinity and bioactivity on NK cell activation, especially E6K [14]. Mature human IL-18 Thr63 may stabilize core structure and loop flexibility. Human, rat, bovine, horse, and mouse IL-18s conserved these properties (Figure 1). This residue was buried in the Tyr52-Thr63 loop, which is critical for binding site II, in the non-binding region [17]. T63A was first reported to increase loop flexibility for binding interaction. It also increased IL-18 activity through IFN-production in single mutants (T63A), double mutants (E6K/T63A and V11I/T73A), and triple mutants (S10T/D17N/T63A) [16,17].

MD simulations showed that T63A increased binding site I loop (Glu31-Thr45) and binding site III loop (Gly108-Lys112) flexibilities without tertiary structure modification (Figure 2 and Figure 3). The 63rd residue substitution also eliminated the IL-18 structure’s strongest H-bond (Ile99@O-Thr63@HG1) to increase structural flexibility (Figure 4F). E6K’s effects at Glu31-Thr45 and Asp142-Met150 loops of binding site I and III controlled E6K/T63A conformational change. Thus, the coupled mutant was more bioactive and suitable for protein structures than its single mutants. Due to the lack of engineered IL-18 structure studies, we performed homology modeling and MD simulation of the ideal E6K/T63X and the single mutant IL-18s (E6A, E6K, T63A, and M33Q). Our simulation time was 150 ns with acceptable RMSD values (Appendix A). Then, binding site I (Glu31-Thr45 loop) showed the high flexibility of most engineered IL-18s. This secondary structure prediction showed that this region formed a loop structure, which was previously reported as a partial α-helix and a full α-helix (Figure 1) [14,15,16]. Glu31 and Asp32 may be flexible residues in the binding groove (Figure 6). However, homology modeling from the NMR structure, PDB code 1J0S, exhibited less loop stability than the crystal structure, PDB code 3WO2. Thus, the helix–loop transition could switch its dynamic structure when penetrating the binding pocket with multistep IL-18 signal complex formation [31,32].

Molecular insight into E6K in this double mutant displayed that Lys53 at binding site II pointed toward Thr299 of IL-18Rα. Then, Asp37, Arg39, Asp40, and Asn41 at binding site I moved closer to the binding pocket. Nevertheless, the obscure effects of the non-binding residue on the engineered IL-18 structure were unclear. Here, we altered the 63rd residue coupling with E6K in order to understand the indirect effects of the non-binding residue substitution using MD simulation, H-bond analysis, and structural analysis. The structural basis of IL-18/IL-18Rα/IL-18Rβ clarified [12,25] that binding site I and II of IL-18 contacted IL-18Rα, whereas binding site III attached to IL-18Rβ (Figure 6).

Here, our MD results could provide the new stage of the binding mechanism. We superimposed the simulated wild type, which represented the free form of IL-18, with the crystal IL-18 of PDB id: 3WO4 structure (Figure 6A). The large binding site I loop (Glu31-Thr45), which was flexible in most mutants, was in the deepest groove of the binding site (Figure 6B). Glu31 and Asp32 formed a coil structure in the simulated IL-18, while Arg13, Asp17, and Asp132 remained in the crystal structure. IL-18 used Arg13, Asp17, and Asp132 to fix the position, then the helix–coil structure penetrated the pocket of binding site I. The distance pattern and RMSF values also affected these regions. When the fixed residues were reversed, the coil structure changed conformation. Second, IL-18 Leu5, Lys53, and Met60 on β-sheets fixed the receptor at binding site I and II (Figure 6C). Lys93’s side chain locked the binding pockets.

Binding site II’s backbone was conserved as in the IL-18 crystal structure, supporting the results obtained from the study on the distance pattern. The two binding site III loops clipped IL-18Rβ (Figure 6D). The non-binding residue alteration affected the open-and-closed structure of this region. The superimposed structure showed the that the closed structure was shaped to interact with the receptor and driven by the Asp142-Met150 loop. To close the structure, simulated IL-18’s Arg147 side chain was bent. His109, Asp110, and Lys112 also connected Val211, Tyr212, Asp213, Tyr214, and Lys313 of IL-18Rβ to maintain its groove. Thus, binding site III structural behaviors showed that the Gly108-Lys112 loop had limited distance patterns and flexibilities compared to the Asp142-Met150 loop.

### 3.2. Role of Hydrogen Bonding of T63X on Conformation Change in Mutant IL-18

The engineered IL-18 structures were transformed into conventional and 63rd residue-associated H-bonds in many regions, except binding site III (Figure 4 and Figure 5). These H-bonds altered binding site III and the 63rd residue cavity. We compared the binding loop direction and Tyr52/Thr73/Ile99 ring in wild type and E6K/T63W structures (Figure 7).

Other engineered IL-18s did not change the cavity size (Appendix A). Three β-sheets (Ala61-Lys67, Ile99-Arg104, and Met113-Ser116) connected Thr63 at the core structure to binding site III (Figure 7A). Ile149@O-Gln114@HE22, Phe102@O-Glu116@H, and Phe101@O-Val62@H conventional H-bonds stabilized these sheets. These bonds conserved at conventional H-bonds in wild type closed two binding site III loops (green box) (Figure 7A). The Met113-Ser116 sheet and binding residues (His109, Asp110, Lys112, Gly145, and Arg147) aligned (Figure 7B).

The Tyr52/Thr73/Ile99 ring was narrow enough to enclose Thr63, the normal IL-18 conformation (Figure 7C). The open structure of binding site III loops in E6K/T63W, which was substituted by the largest amino acid at the 63rd residue, differed from the wild type (Figure 7D). To expose the surface protein, the binding residues encircled the Met113-Ser116 sheet (Figure 7E). Ile149@O-Gln114@HE22 bond reduced occupancy from 83.1% (wild type) to 16.8% (E6K/T63W) in the open loop. When Ile99 escaped from the substituted residue, it enlarged the Tyr52/Thr73/Ile99 ring around Trp63 (Figure 7F). To maintain the 63rd residue cavity without distracting Tyr52 and Thr73 for H-bond formation, Ile99 and Trp63’s side chain had 61.1% H-bond occupancy.

Evidently, H-bond formation can regulate the IL-18 mutation by changing conformation. We found the locked arms at binding sites I and II and the clip of the open/close structure at binding site III (Figure 7A). The large binding site I interface between IL-18 and IL-18Rα had a surface area of about 1850 Å^2^. The helix–coil transition may move like an arm to fit in the binding groove of various IL-18 helix architectures (Figure 6B). Like binding site II, the Lys93 arm of IL-18 held onto the IL-18Rα acidic surface and captured Lys53 (Figure 6C). Two binding site III loops clipped IL-18Rβ (Figure 6D) while the lateral side recognized IL-18/IL-18Rα.

The non-binding residue’s 63rd residue alteration could change its cavity by H-bonding with Ser50 and Tyr52/Thr73/Ile99 ring (Figure 5B and Figure 7C). The side chain of the 63rd residue was connected to Ile99 of the Ile99-Arg104 sheet, while its backbone maintained two core structure β-sheets with Ser50. The side chain of the 63rd residue formed H-bonds with Tyr52 and Thr73 after the substitution. Pro89@O-Tyr52@HH and Met86@O-Thr73@HG1 also maintained the Glu85-Ile100 loop lid of the 63rd residue (Figure 4E). Thus, large-residue substitution may structurally disrupt ring and lid preservation.

The Ile49/Met51/Val62 ring near this cavity may be important for IL-18-virus IL-18BP binding (Figure 5A) [22,29]. First, IL-18 Met51@S formed a pi–sulfide bond with *Ectromelia virus* (Ectv) IL-18BP binding site B Tyr51@O. Second, the hydrophobic wall of IL-18 (Lys8, Met51, Met60, Gln103, Met113, and Val153) formed extensively with EctvIL-18BP binding site. Thirdly, IL-18 Ile49 and Val62 formed an aliphatic hydrophobic wall floor. The conserved Met51@O-Leu5@H and Phe101@O-Val62@H bonds may protect the wall and floor structures from the 63rd residue alteration (Figure 4F). AlphaFold, an AI system, predicted the human IL-18BP structure (AF-095998-F1) [33].

### 3.3. Conclusive Remarks of Thr63 Alteration Towards IL-18 Atomistic Information

To clarify the impact of the Thr63 mutation on IL-18 dynamics in IL-18 improvement, Table 1 depicted the conclusive effects caused by the numerous mutations. This study looked at the effects on conformational change and hydrogen bond occupancy of IL-18 binding sites when compared to wild type IL-18. The E6K/T63A was highlighted because the experiment revealed that this engineered IL-18 is more active through IFN production than the wild type [16,17]. Only sites I and II showed hydrogen-bond occupancy, while site III had no conventional H-bonds. The Glu85-Ile100 loop, which is located near Thr63, was also included. Furthermore, the E6K/T63K alteration was excluded due to its inability to be energy-minimized, as previously mentioned in this study.

According to Table 1, no change in conformation at binding site II was observed as a result of Thr63 alteration; however, E6K primarily contributed to the change in conformation at binding site I, while both E6K and T63 alterations accounted for the changes at binding sites I and III in the majority of cases. Only E6K/T63V showed no change in conformation at both sites.

Apart from the conformational aspect, it is possible that the Thr63 alteration affected hydrogen bond occupancy at binding site I and the Glu85-Ile100 loop, but not binding site II. The same result was obtained for E6K and E6K/T63A, leading to speculation that the change in hydrogen bond occupancy may not be sufficient to investigate the relationship between the replaced amino acid at position 63 and IL-18 activity. Instead, the conformational change could serve as a design guideline for IL-18.

E6K and E6K/T63A, as well as M33Q, contained the conformational change at binding sites I and III. These IL-18 mutants were found to have a change in IL-18 activity when compared to WT IL-18 in experiments. These findings could indicate that E6K/T63X, which can cause changes in composition at binding sites I and III, could be viable options for developing new effective engineered IL-18.

### 3.4. Limitation of IL-18 Atomistic Studies

For structure-based drug development, murine DR-18 with N1H/M50A/K52G/E55R/V56A/L59K mutation could be useful for analyzing the interactions occurring between human DR-18 and IL-18BP. In this case, we can determine the indirect effects of the non-binding residue on the binding residue interaction in the IL-18 structure. Several study designs and computational methods found that promoting a conformational switch by non-binding residue was interpreted for IL-2 and IL-1. As a result, this concept could be extended to other engineered ILs or cytokines.

However, structural analysis of the engineered IL-18 and its complex structure is required to better understand the binding affinity. This study has several limitations which need to be addressed: (a) Because this in silico study is purely computational and relies on experimental structure analysis, the crystal structure of the mutant/engineered IL-18 is required for validation. (b) Protein expression and purification of the engineered IL-18 should be performed to ensure stability. (c) The activity of the proposed engineered IL-18 should be tested to ensure that the experimental and computational results are consistent.

## 4. Materials and Methods

### 4.1. Sequence Alignment

The secondary structure of mature human IL-18 was predicted by PSIPRED version 4.0 webserver [34]. Sequence alignment of mammalian IL-18s was accessed via Uniprot databank on 12 July 2021. The full IL-18 protein sequences were based on the accession numbers Q14116 (human), P97636 (rat), Q9TU73 (bovine), Q9XSQ7 (horse), and P70380 (mouse), respectively.

### 4.2. Molecular Dynamics Simulation

Homology modeling of engineered IL-18s was based on a mature human IL-18 amino acid sequence (Uniprot: Q14116). T63 was altered to other amino acids coupled with E6K using Phyre2 [24] to create the initial conformer of the human IL-18 (PDB id: 1J0S) [12]. The protein models were listed in Appendix A. The superimposed structures and the template demonstrated the identical protein backbones by RMSD values. The protonation states of amino acid side chains were adjusted at pH 7.0 based on Poisson–Boltzmann electrostatics calculation by the PDB2PQR web server [35].

All systems were prepared using the tleap module in the Assisted Model Building with Energy Refinement (AMBER) version 16 software package [36,37] with a condition of 14 Å of TIP3P water box [38] and 150 mM NaCl. The protein solution contained approximately 12,200 to 12,400 water molecules and 33 NaCl pairs. Energy minimization of the solvated protein was computed with 1000 steps of the steepest descent and 1000 steps of the conjugate gradient under a periodic boundary condition by the Particle Mesh Ewald Molecular Dynamics (PMEMD) module [39] in AMBER16. Unfortunately, E6K/T63K was unable to minimize due to a high van der Waals force from the lysine sidechain. Therefore, this mutant was neglected for MD simulation.

An isothermal ensemble (NVT) was performed to equilibrate the protein solution system using Langevin dynamics at 310 K. The harmonic potential was assigned to all protein atoms with the respective force constants (200, 100, 50, 20, and 10 kcal/mol-Å^2^) and a 1 fs of time step [16]. The bonds involving hydrogen were constrained using the SHAKE algorithm [40]. The system was then converted to an isobaric/isothermal ensemble (NPT) with a 2 fs of time step using the weak coupling algorithm [41]. The atomistic simulation was investigated at 310 K and 1.013 bar (1 atm) for 150 ns with a nonbonded cutoff of 16 Å and the periodic boundary condition, similar to previous studies [42,43,44]. The first 90 ns of the simulation were omitted, and the last 60 ns were collected as 1500 equidistant snapshots for a configuration average and analysis. All simulated proteins during 90–150 ns reached equilibrium by the stable RMSD value (Appendix A). The acceptable average value was less than 4 Å. Because the RMSD was well converged, as in the previous study [45], the simulation was performed in a single 150ns simulation with no additional replicates.

### 4.3. Structural Analysis

The effects of T63 alteration coupled with E6K on IL-18 were investigated by the following parameters. All IL-18 display and analysis was conducted using the Visual Molecular Dynamics (VMD) program, version 1.9.3 [46] and the CPPTRAJ module [47] in AMBER16. RMSD and RMSF calculations were based on alpha carbon atoms of the proteins. The overall conformational change was analyzed by the distances found between the center of the 63rd residue and other residues at the backbone atoms (N, C, and Cα), referred to as the center of geometry.

Intramolecular H-bond analysis tracked all solute–solute H-bonds and followed the default setting of the angle and distance cutoff, 135 degree and 3.0 Å, respectively. An interaction with higher than 80% occupancy was assumed to be a conventional H-bond. The summation of hydrophobicity index was calculated using the Eisenberg scale [48]. The amino acid with the more positive index value was considered to be more hydrophobic, whereas the hydrophilic property was contrasted with the negative value.

## 5. Conclusions

Engineered ILs have emerged as a promising strategy for a wide range of therapeutics, particularly cancer immunotherapy [49]. IL-18-based therapies have also shown promise in preclinical and clinical studies [50], including cytokine-induced memory-like NK cell therapy [51,52], IL-18-secreting CAR T cell therapy [5,6,7], combination therapies of IL-18 and immunostimulatory cytokine [53,54], and combination therapies of IL-18 and targeted therapy [4,55,56]. Furthermore, many engineered ILs, such as IL-2, IL-10, IL-18, and IL-23, had their functions established through in vitro, in vivo, and in silico studies [4,18,21,57,58].

In conclusion, the indirect effects of 63rd residue alteration coupling with the E6K binding residue played important roles in IL-18 conformational change and flexibility. We have provided perspectives on structure-based IL-18 engineering, including binding site behaviors, essential H-bonds of conformational switches, and atomistic insights into protein regions. As a result, interdisciplinary cytokine engineering research could be a key goal for the next generation of treatment, and expanding our understanding of IL structure could be beneficial for cytokine engineering, which will be a powerful treatment strategy in the future.

## Figures and Tables

**Figure 1 ijms-25-12992-f001:**
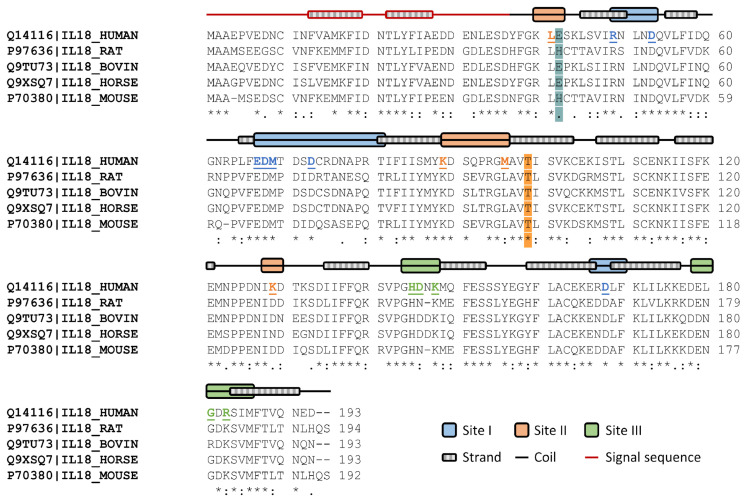
Mammalian IL-18 protein sequence alignment. Human, rat, bovine, horse, and mouse IL-18 protein sequences were obtained from the Uniprot database. The residues highlighted in cyan represent glutamic acid at the 42nd and 6th residues of full human IL-18 and mature human IL-18, respectively. The residues highlighted in orange are threonine residues, specifically the 99th residue in the full sequence and the 63rd in the mature sequence. The signal and mature sequences are represented by the red and black lines, respectively. The color boxes for the binding regions were based on the mature sequence, which included binding sites I, II, and III. Each binding site’s bold and underlined amino acids are human IL-18 receptor-interacting residues. The fully conserved, strongly similar, and weakly similar residues are labeled by asterisks (*), colons (:), and periods (.), respectively.

**Figure 2 ijms-25-12992-f002:**
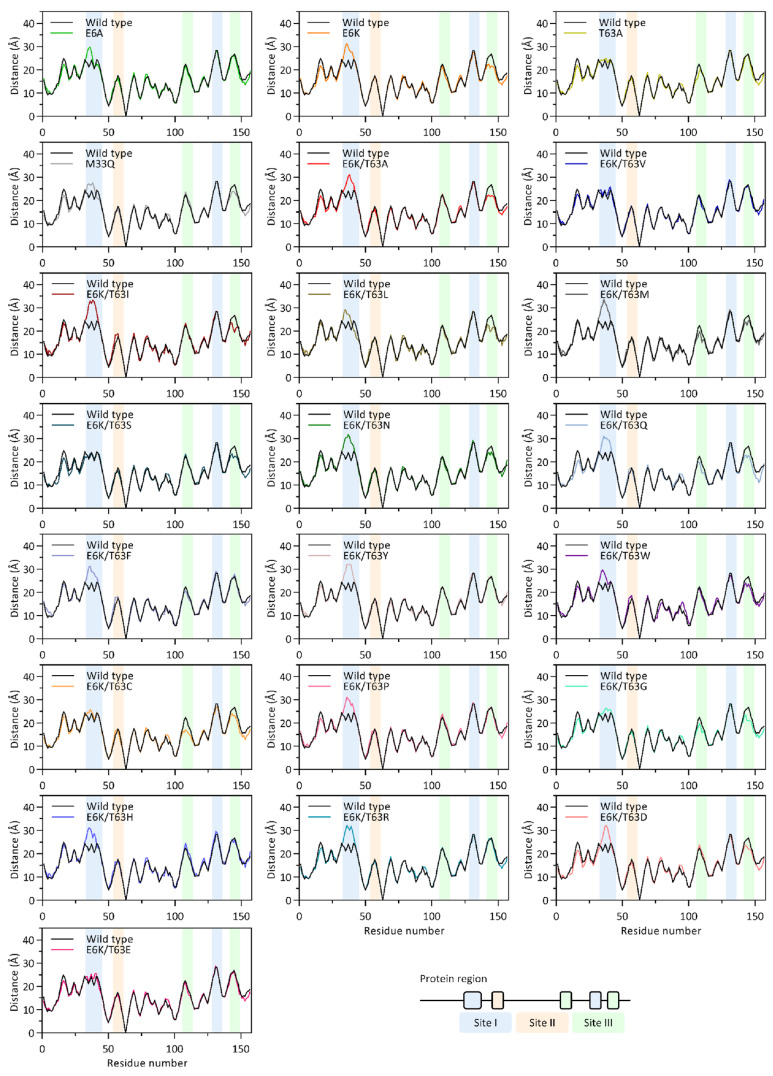
Distance pattern of E6K/T63X. The distance pattern was calculated between the 63rd residue and the residue of interest. Wild type IL-18 acts as a baseline (black). The blue, orange, and green bars represent the regions of binding site I, II, and III, respectively.

**Figure 3 ijms-25-12992-f003:**
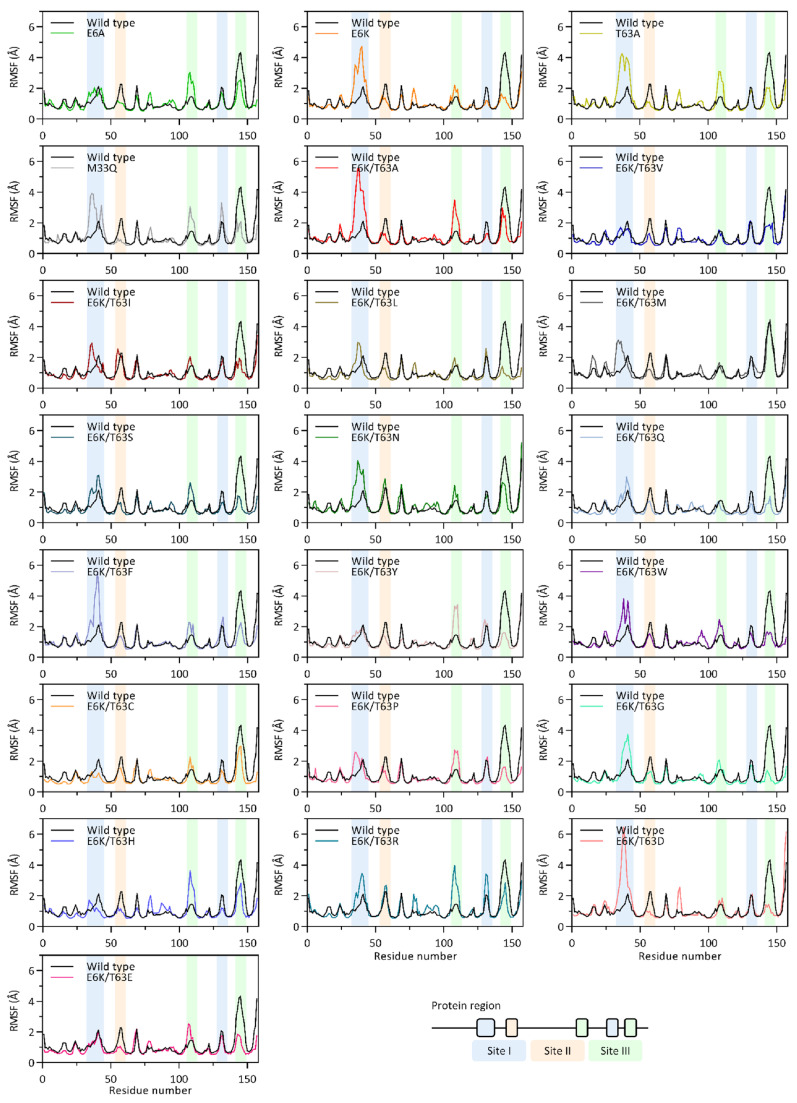
Root-mean square fluctuation of E6K/T63X. All graphs have the same illustration as Figure 2.

**Figure 4 ijms-25-12992-f004:**
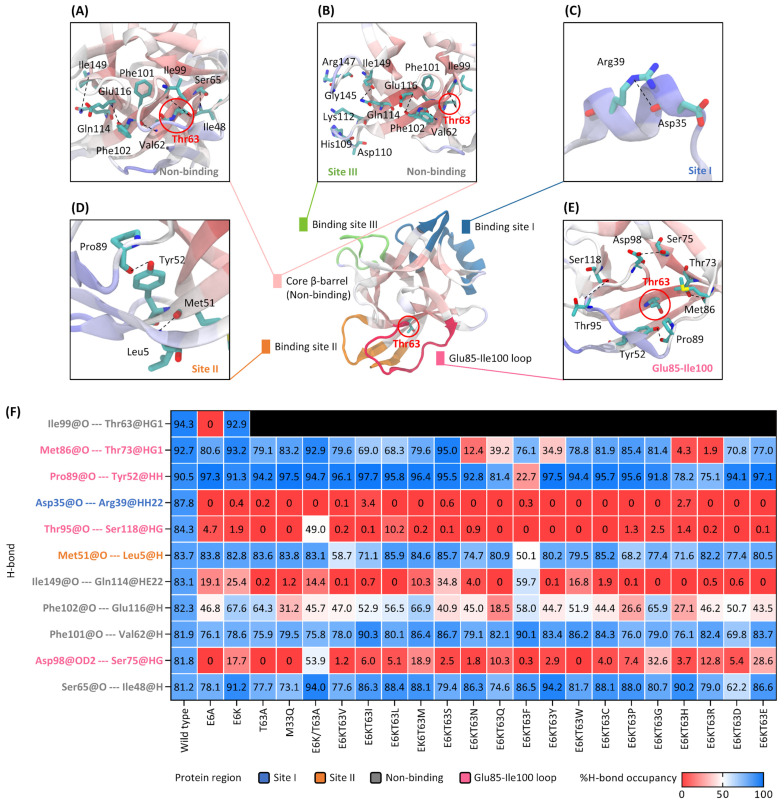
IL-18 conventional H-bond analysis. Structural insight of the 63rd residue and conventional H-bonds at non-binding region (**A**), binding site III and non-binding region (**B**), binding site I (**C**), binding site II (**D**), and Glu85-Ile100 loop (**E**). Dashed lines illustrate the conventional H-bonds from the donor to the acceptor atoms. (**F**) Conventional H-bond occupancies of E6K/T63X compared to wild type IL-18. The colored H-bonds at the y-axis indicate the H-bond location, including the non-binding region (gray), the Glu85-Ile100 loop (pink), binding site I or helix (blue), and binding site II (orange). However, binding site III prevented conventional H-bond analysis. The H-bond naming system of the donor (oxygen) or acceptor (hydrogen) atom is presented as an amino acid, a residue number, and an oxygen atom or a hydrogen atom. The donor/acceptor atom of the side chain is indicated with the order (H: alpha, G: gamma, D: delta, and E: epsilon) and/or the number. The %H-bond occupancy shade from 0% (red), 50% (white), and 100% (blue). The black box indicates the unavailable H-bond.

**Figure 5 ijms-25-12992-f005:**
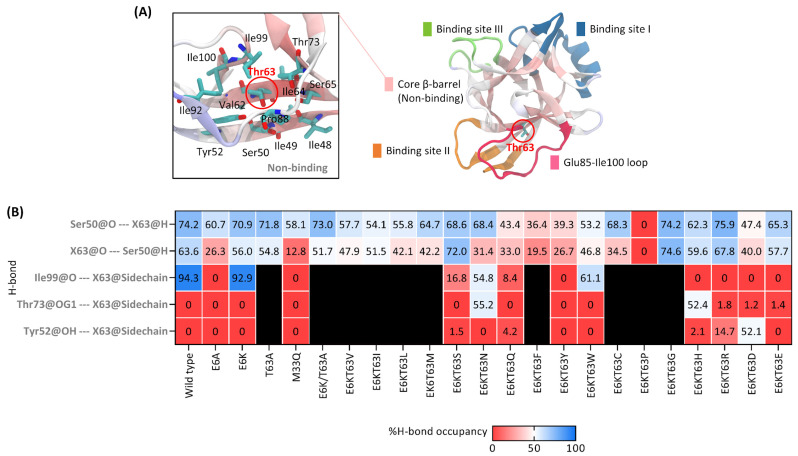
H-bond analysis of the 63rd residue-associated bonds. (**A**) Structural insight of T63 and neighbor residues within 5 Å around the 63rd residue at core β-barrel and Glu85-Ile100 loop. The hydrophobic residues of this region are Ile48, Ile49, Val62, Ile64, Pro88, Ile92, Ile99, and Ile100, whereas the hydrophilic residues are Ser50, Tyr52, Ser65, and Thr73. (**B**) H-bond occupancies of the 63rd residue-associated bonds. The backbones of the 63rd residue and Ser50 of wild type had moderate H-bond occupancies. The side chain of the 63rd residue can interact with Tyr52, Thr73, and Ile99. X63 and X63@Sidechain indicate the 63rd residue alteration and its side chain. All illustrations have the same label as Figure 4. The black area denoted the area is not applicable to H-bond.

**Figure 6 ijms-25-12992-f006:**
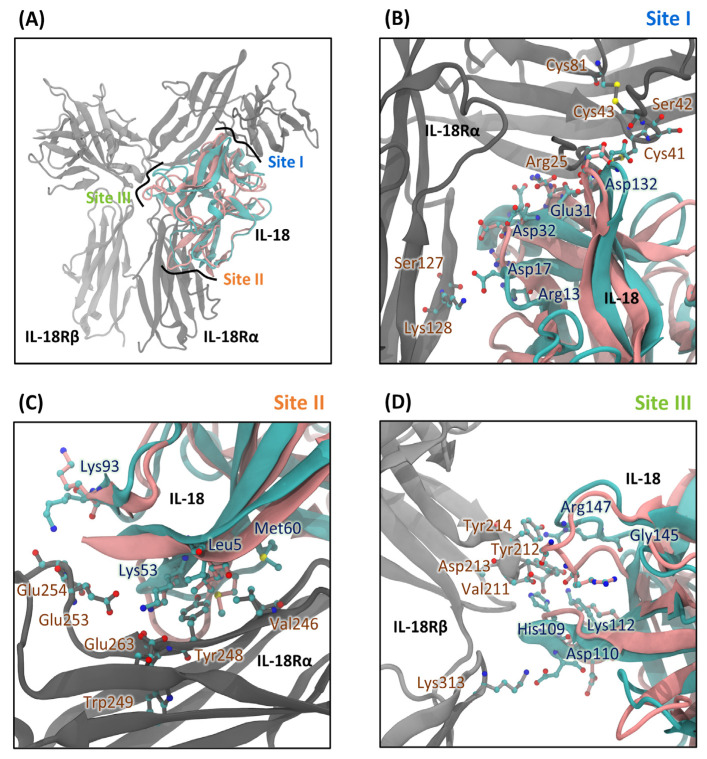
Structural insight of IL-18/IL-18Rα/IL-18Rβ signaling complex. (**A**) Ternary complex of wild type IL-18 and IL-18 receptors. The simulated IL-18 (salmon) and the crystal structure of IL-18 (cyan) from 3WO4 were superimposed. The binding sites of IL-18 to IL-18Rα (dark gray) and IL-18Rβ (light gray), including binding sites I, II, and III, were illustrated. The binding residues were aimed at the binding pockets containing the apo form of simulated IL-18, which bent the binding side chains in various directions. This was particularly true of Asp32 of binding site I (**B**), Lys93 of binding site II (**C**), and Asp110/Arg147 of binding site III (**D**).

**Figure 7 ijms-25-12992-f007:**
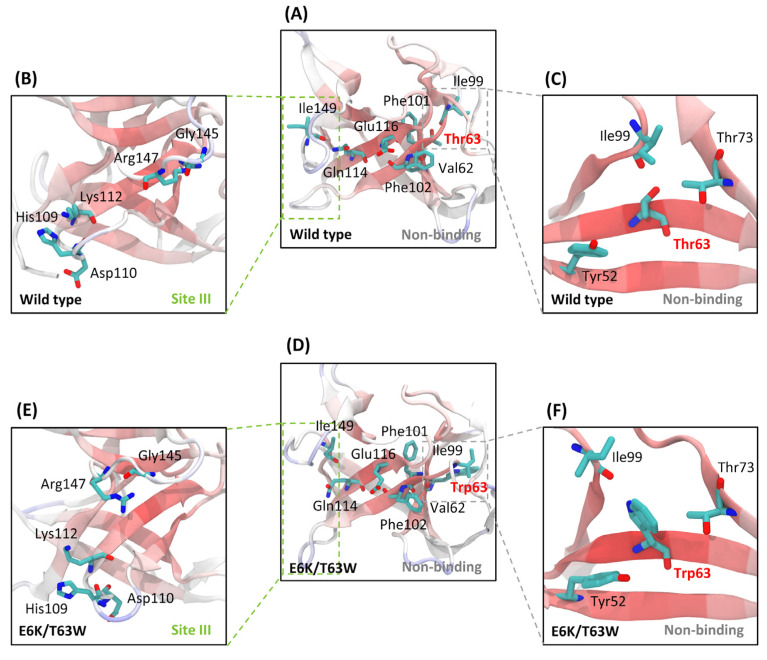
The effects of the 63rd residue cavity modification. Changes in the wild type and E6K/T63W structures were explored at the non-binding region (**A**,**D**), binding site III (**B**,**E**), and the 63rd residue cavity (**C**,**F**). Binding site III and the 63rd residue region were linked by Val62, Phe101, Phe102, Glu116, Gln114, and Ile149, which involved conventional H-bonds of the non-binding region. While binding site III loops of E6K/T63W were shifted from wild type, the 63rd residue cavity of E6K/T63W (Trp63/Thr73/Ile99) was larger than the wild type (Thr63/Thr73/Ile99).

**Table 1 ijms-25-12992-t001:** Summary of the change in conformation and hydrogen-bond occupancy of mutant Il-18 in a comparison with WT IL-18.

Alteration	Change in Conformation of IL-18 Binding Site, Compared to WT IL-18			Change in Hydrogen-Bond Occupancy of IL-18 Binding Sites, Compared to WT		
	I	II	III	I	II	Glu85-Ile100 Loop
WT	-	-	-	-	-	-
E6A	X	X	X	√	X	√
E6K	√	X	√	√	X	√
M33Q	√	X	√	√	X	√
T63A	X	X	√	√	X	√
E6K/T63A	√	X	√	√	X	√
E6K/T63C	√	X	√	√	X	√
E6K/T63D	√	X	√	√	X	√
E6K/T63E	√	X	X	√	X	√
E6K/T63F	√	X	√	√	√	√
E6K/T63G	√	X	√	√	X	√
E6K/T63H	√	X	X	√	X	√
E6K/T63I	√	X	√	√	X	√
E6K/T63L	√	X	√	√	X	√
E6K/T63M	√	X	√	√	X	√
E6K/T63N	√	X	√	√	X	√
E6K/T63P	√	X	X	√	X	√
E6K/T63Q	√	X	√	√	X	√
E6K/T63R	√	X	X	√	X	√
E6K/T63S	X	X	√	√	X	√
E6K/T63V	X	X	X	√	√	√
E6K/T63W	√	X	√	√	X	√
E6K/T63Y	√	X	X	√	X	√

Note: The symbols “I”, “II”, and “III” represent IL-18 binding sites I, II and III, respectively. The symbol “√” and “X” denote the distinctive alteration or lack of alteration, respectively. The distinctive change in the conformation (symbol “√”) at the respective binding site is defined by any changes in distances of more than 2 Å between the specified alteration and WT IL-18, according to the distance patterns. A distinctive change in the hydrogen-bond occupancy (symbol “√”) is defined as a difference of more than 20% between the interested alteration and WT. E6K and E6K/T63A are highlighted in yellow, since the experiment showed that this engineered IL-18 exhibits greater activity through IFN-production than the wild type [16,17].

## Data Availability

Data is contained within the article and Appendix A.

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
