# Peer review of "Role of Non-Binding T63 Alteration in IL-18 Binding"

_ijms, 2024, doi:10.3390/ijms252312992_

Round 1
Reviewer 1 Report (New Reviewer)
Comments and Suggestions for Authors
The paper encoded ijms-3091204 submitted by C. Peeyatu & colleagues is a potentially interesting contribution to unveil the molecular determinants underlying the interactions and binding of interleukines with their targets, in particular of IL-18.
However, the manuscript presents many weakness points that impede it to be published as it is. Hence, a strong, MAJOR revision is required!
The main issue that clearly emerges is related to many English style and grammar errors, as well as sentence constructions that make the reading and the understanding very difficult. Then, also the titles of paragraphs 2.2 and 2.2.2 are totally confusing.
In general, the REVISION MUST be helped, in my opinion, by a native English contribution.
The study is presented almost as a list of effects of many, really numerous, structural and dynamics effects induced by the mutations in the various IL-18 loops. These effects seem to overpass the effects of the mutations of the T63 residue alone. Again, in the manuscript flowing, it is sometimes not clearly explained if the T63 mutations are analysed together with E6K or not, and/or in the context of a double or triple mutants…
Then, the results are sometimes accompanied by a partial discussion, that is turn deepened in the Discussion Section. I know that these two aspects are difficult to be precisely divided, but an idea could be to make a unique Results and Discussion Section with a final paragraph that discusses all the principal results obtained and their rebounds for the aim of the study.
Hence, since the final purpose of the study is to clarify the role of IL-18 dynamics in a immunotherapeutic treatment strategy especially against cancer, I strongly suggest to insert a Table in which the different type of effects induced by the numerous mutations introduced are clearly correlated to the actual series of mutations present in each particular mutant.
In detail, some issues to be amiliorated:
1-INTRODUCTION: it is not clearly explained the difference existing within the IL1 and IL18 family (2nd paragraph of the Intro). Even a short sentence could be spent to better clarify this aspect. Something is then reprised in the Discussion concerning IL-37. Why IL-37 is included in the IL-18 family if it is a dimer? (Ref 33). Some clarifications could be introduced to better ease the interpretation of the meaning of the study.
The sentence in the 16th and 17th lines of the INTRO must be amended, as well as the lines 25-27 (page 2). As well, the first line in page 3 must be better written.
2-RESULTS: page 3, 4 lines from the bottom: amend was with “is” (after Glu6), and check similar cases throughout the whole manuscript. Again, the following sentence, MUST be amended as “On the other hand, a Thr in position 63 was highly conserved in the sequence of mature IL-18 in all organisms.” Similar awfully presented sentences appear throughout the whole manuscript…
2-RESULTS, page 4: the whole 2.1.2 paragraph MUST BE COMPLETELY RE-WRITTEN, like the first three lines of 2.13 paragraph. Then, 9 lines from the bottom: what means “CONCEPT”??!! Did the authors mean “results”?!...
2-RESULTS, page 7: “in a manner same as a wild type”… is badly written!!
Then, last line of the paragraph (middle of the page): amend the sentence as “This loop becomes more rigid in the E6K/T63K mutant as a result of the conformational change occurring in this region”.
Paragraph 2.2.1: what is the meaning of the second sentence?!? It is not clear at all. Then, amend Ile199 with Ile99.
2-RESULTS, page 9, middle of 2.2.2. paragraph: please clarify the meaning of the sentence starting with “As previously stated…”
3-DISCUSSION, page 11: the two 1J0S and 3WO2 codes must be reported as pdb codes. Then, delete “was” before ‘clarified’ in the sentence preceding Figure 6. Again, add a “comma” before ‘whereas’. Finally, add “of” between ‘form’ and ‘IL-18’ two lines from the bottom of the page.
3-DISCUSSION, page 12: amend ‘as the crystal IL-18…’ with “as in the IL-18 crystal structure, supporting the results obtained from the study of the distance(s) pattern”. Then, re-write also the sentence before Figure 7!
3-DISCUSSION, page 13, 16 lines from the bottom: amend with “… mutations could be useful to analyze the interactions occurring between human DR-18 and IL-18BP”.
I mean, the mutation(s) cannot study anything! They only have the role to suggest a study or and analysis!
5: CONCLUSIONS, page 15: a sort of “Conclusions” are better written at the beginning of page 14 (last part of the Discussion)! Hence, the authors can manage these two parts to provide a more efficacious (not repetitive) underlining of the value of the study.
Many other sentences could be ameliorated following the indications I reported above...

Author Response
Reviewer 1
General aspect
- In general, the REVISION MUST be helped, in my opinion, by a native English contribution.
Response:
We have sent the manuscript for the proofreading and language editing to the native English editor from Office of International Affairs, Faculty of Medicine, Prince of Songkla University. We also stated this in the acknowledgement section as “V.T. would also like to express a gratitude to Office of International Affairs, Faculty of Medicine, Prince of Songkla University for editing the revised manuscript.”.
- Then, the results are sometimes accompanied by a partial discussion, that is turn deepened in the Discussion Section. I know that these two aspects are difficult to be precisely divided, but an idea could be to make a unique Results and Discussion Section with a final paragraph that discusses all the principal results obtained and their rebounds for the aim of the study.
Response:
We have divided the discussion into three sections in the revised manuscript:
3.1 Binding site analysis of wild type and engineered IL-18 via MD trajectories
3.2 Role of hydrogen bonding of T63X on conformation change of mutant IL-18s
3.3 Limitation of this IL-18 atomistic studies
- Hence, since the final purpose of the study is to clarify the role of IL-18 dynamics in a immunotherapeutic treatment strategy especially against cancer, I strongly suggest to insert a Table in which the different type of effects induced by the numerous mutations introduced are clearly correlated to the actual series of mutations present in each particular mutant.
Response:
We have added the additional section in discussion part. The Table 1 was done to illustrate the effect of the T63 alteration on the simulated IL-18 structures, compared to WT.
Introduction
1) it is not clearly explained the difference existing within the IL1 and IL18 family (2nd paragraph of the Intro). Even a short sentence could be spent to better clarify this aspect. Something is then reprised in the Discussion concerning IL-37. Why IL-37 is included in the IL-18 family if it is a dimer? (Ref 33). Some clarifications could be introduced to better ease the interpretation of the meaning of the study.
Response:
Thanks for the suggestion. First of all, to avoid the confusion and unclear explanation, we have focused on only IL-18 detail in the 2nd paragraph instead by using the sentence “IL-18 plays important roles in immune homeostasis [9], and its functions have been identified as pro-inflammatory cytokines [10–12].” instead of stating the IL-1 along with IL-18.
For the point of the IL-37, at first, we have included this IL-37 to this writing as IL-37 could bind IL-18R, however, this could make unclear statement and irrelevant overclaim, or exaggeration for this study so we decide to remove this claim or speculation about IL-37. To prove the statement and discussion, the study of IL-37 is necessary. Therefore, we then only discuss only IL-18 and state the limitation of the study as followings:
“In this case, we can determine the indirect effects of the non-binding residue on the binding residue interaction in the IL-18 structure. Several study designs and computational methods found that promoting conformational switch by non-binding residue was interpreted for IL-2 and IL-1. As a result, this concept could be extended to other engineered ILs or cytokines. However, structural analysis of the engineered IL-18 and its complex structure is required to better understand the binding affinity. Some limitations in this study need to be addressed: a) Because this in silico study is purely computational and relies on experimental structure analysis, the crystal structure of the mutant/engineered IL-18 was required for validation. b) Protein expression and purification of the engineered IL-18 should be performed to ensure stability. c) The activity of the proposed engineered IL-18 should be tested to ensure that experimental and computational results are consistent.”.
2) The sentence in the 16th and 17th lines of the INTRO must be amended, as well as the lines 25-27 (page 2). As well, the first line in page 3 must be better written.
Response:
We amended the 16th and 17th lines as “IL-18 plays important roles in immune homeostasis [9], and its functions have been identified as pro-inflammatory cytokines [10–12]. Site-directed mutagenesis could be used to identify the critical amino acid residue that influences IL-18 function. The 6th glutamic acid (E6) and the 53rd lysine (K53) are important amino acids for IL-18R and IL-18BP binding [13,14]. Changing E6 to lysine (E6K) improves NK cell activation protein function [14,16], while changing K53 to glutamic acid (K53E) reduces cytokine activity [16]. In contrast, some mutated IL-18s, D17N and M33Q, altered the stabilization of site I, resulting in a decrease in IFN-γ [15,16].”
The lines 25-27 in page 2 and their related contents were edited to “Previous studies have used molecular dynamics (MD) simulation to investigate the structure of IL-18 [16,22,23]. The stability of the IL-18s, IL-18/IL-18R, and IL-18/IL-18BP was evaluated so that they could be used in structure-based molecular drug design [22]. Prior studies found that E6K/T63A mutations in the IL-18 structure influenced structural flexibility and interaction [16]. However, how the 63rd residue affects protein structure is unknown.”.
The first line in page 3 and their related contents were edited to “The full IL-18 protein sequences were used to compare the similarity of E6 and T63 across species. Protein sequences were obtained from the Uniprot database for humans (Q14116), rats (P97636), cattle (Q9TU73), horses (Q9XSQ7), and mice (P70380) (Figure 1).
The E6 is present in humans, cattle, and horses, while rats and mice sequences contain histidine at this location. Despite being in the coil of binding site II (K4-E6), E6 was not a binding residue. T63 is conserved in all selected organisms. The identity percentage of full-length IL-18 sequences in rats, bovines, horses, and mice was 62.1%, 77.7%, 79.3%, and 62.3%. Table S1 shows the results of the protein sequence alignment analysis.”.
Results:
1) page 3, 4 lines from the bottom: amend was with “is” (after Glu6), and check similar cases throughout the whole manuscript. Again, the following sentence, MUST be amended as “On the other hand, a Thr in position 63 was highly conserved in the sequence of mature IL-18 in all organisms.” Similar awfully presented sentences appear throughout the whole manuscript…
Response:
Thanks for suggestion. In page 3, these 4 lines were edited to “The E6 is present in humans, cattle, and horses, while rats and mice sequences contain histidine at this location. Despite being in the coil of binding site II (K4-E6), E6 was not a binding residue. T63 is conserved in all selected organisms. The identity percentage of full-length IL-18 sequences in rats, bovines, horses, and mice was 62.1%, 77.7%, 79.3%, and 62.3%. Table S1 shows the results of the protein sequence alignment analysis.”.
2) page 4: the whole 2.1.2 paragraph MUST BE COMPLETELY RE-WRITTEN, like the first three lines of 2.13 paragraph. Then, 9 lines from the bottom: what means “CONCEPT”??!! Did the authors mean “results”?!...
Response:
In the revised manuscript, the whole 2.1.2 paragraph was rewritten as well as the first three lines of 2.13 paragraph. The “concept” means “results”, so we have rewritten into “This binding loop flexibility result could be related to the structural effects of E6K and T63A on the coupled mutant.”.
3) page 7: “in a manner same as a wild type”… is badly written!!
Then, last line of the paragraph (middle of the page): amend the sentence as “This loop becomes more rigid in the E6K/T63K mutant as a result of the conformational change occurring in this region”.
Response:
In the page 7, we changed the sentence into “In terms of binding site II flexibility, most engineered IL-18s had lower RMSF values than wild type (Figure 3). E6K/T63I, E6K/T63N, and E6K/T63R, on the other hand, conserved the region's flexibility in the same way that the wild type did.”. We also amended the sentence as “This loop becomes more rigid in the E6K/T63K mutant as a result of the conformational change occurring in this region”. Thanks very much.
4) Paragraph 2.2.1: what is the meaning of the second sentence?!? It is not clear at all. Then, amend Ile199 with Ile99.
Response:
Thanks for the suggestion. The second sentence was changed into “This H-bond formation and deformation can be a useful parameter for tracking secondary structure changes in each region during the dynamics condition [25].”. Also, “Ile199” was changed into “Ile99”.
5) page 9, middle of 2.2.2. paragraph: please clarify the meaning of the sentence starting with “As previously stated…”
Response:
We change the word “As previously stated” to be “In addition to the Thr63 backbone-related H-bond” to avoid the ambiguity. We try to mentioned the H-bond from the Thr63 backbone.
Discussion
1) page 11: the two 1J0S and 3WO2 codes must be reported as pdb codes. Then, delete “was” before ‘clarified’ in the sentence preceding Figure 6. Again, add a “comma” before ‘whereas’. Finally, add “of” between ‘form’ and ‘IL-18’ two lines from the bottom of the page.
Response:
Thanks for the suggestion. We have replaced the sentence to “However, homology modeling from the NMR structure, PDB code 1J0S, which had less loop stability than the crystal structure, PDB code 3WO2.” We also corrected a “comma” before ‘whereas’ and sue the “form of IL-18” as suggested.
2) page 12: amend ‘as the crystal IL-18…’ with “as in the IL-18 crystal structure, supporting the results obtained from the study of the distance(s) pattern”. Then, re-write also the sentence before Figure 7!
Response:
Thanks very much. We have changed the sentence as suggested to be “The binding site II backbone was conserved as in the IL-18 crystal structure, supporting the results obtained from the study of the distance(s) pattern.”. For the sentence before Figure 7, we have changed into “We compared the binding loop direction and Tyr52/Thr73/Ile99 ring in wild-type and E6K/T63W structures (Figure 7).”.
3) page 13, 16 lines from the bottom: amend with “… mutations could be useful to analyze the interactions occurring between human DR-18 and IL-18BP”. I mean, the mutation(s) cannot study anything! They only have the role to suggest a study or and analysis!
Response:
Thanks very much for the correction. We do agree and appreciate your guidance. We have amended this sentence into “For structure-based drug development, murine DR-18 with N1H/M50A/K52G/E55R/V56A/L59K mutation could be useful to analyze the interactions occurring between human DR-18 and IL-18BP.”, as suggested.
Conclusion
- page 15: a sort of “Conclusions” are better written at the beginning of page 14 (last part of the Discussion)! Hence, the authors can manage these two parts to provide a more efficacious (not repetitive) underlining of the value of the study.
Response:
We have moved the beginning of page 14 and we rewrote the conclusion as “Engineered ILs have emerged as a promising strategy for a wide range of therapeutics, particularly cancer immunotherapy [49]. IL-18-based therapies have also shown promise in preclinical and clinical studies [50], including cytokine-induced memory-like NK cell therapy [51,52], IL-18 secreting CAR T cell therapy [5-7], combination therapies of IL-18 and immunostimulatory cytokine [53,54], and combination therapies of IL-18 and targeted therapy [4,55,56]. Furthermore, many engineered ILs, such as IL-2, IL-10, IL-18, and IL-23, had their functions established through in vitro, in vivo, and in silico studies [4,18,21,57,58].
In conclusion, the indirect effects of 63rd residue alteration coupling with the E6K binding residue played important roles in IL-18 conformational change and flexibility. We provided perspectives on structure-based IL-18 engineering, including binding site behaviors, essential H-bonds of conformational switches, and atomistic insights into protein regions. As a result, interdisciplinary cytokine engineering research could be a key goal for the next generation of treatment, and expanding our understanding of IL structure could be beneficial for cytokine engineering, which will be a powerful treatment strategy in the future.”.

Reviewer 2 Report (New Reviewer)
Comments and Suggestions for Authors
The manuscript from Peeyatu et al, demonstrates the roles of non-binding T63 alteration on IL-18 binding. The minor concerns are below.
· Mention the full form of Th1, NK, NKT, B, DC.
· “homeostasis in a variety of contexts [9]”. Authors may need to explain it instead of providing broad terms like “variety of contexts”.
· “E6K increases NK cell activation protein function, whereas K53E decreases cytokine activity.” Citation required
· “Few studies had used molecular dynamics (MD) simulation to analyze the structure of IL-18.” Need to cite what are those few studies
· “which was previously reported as a partial α-helix and a full α-helix” citation required.
· Please specify the limitations of the study in the discussion section.
Author Response
The manuscript from Peeyatu et al, demonstrates the roles of non-binding T63 alteration on IL-18 binding. The minor concerns are below.
- Mention the full form of Th1, NK, NKT, B, DC.
Response: We have edited these into “Th1 cells, natural killer (NK) cells, natural killer T (NKT) cells, B cells, dendritic cells (DC), and macrophages” as suggested.
- “homeostasis in a variety of contexts [9]”. Authors may need to explain it instead of providing broad terms like “variety of contexts”.
Response: Thanks for your suggestion. We have removed the word “variety of contexts” to avoid the confusion. We only used the sentence “The IL-1 family and IL-18 play significant roles for immune homeostasis [9].”, which is suffice for the information we would like to mention.
- “E6K increases NK cell activation protein function, whereas K53E decreases cytokine activity.” Citation required
Response: We have edited into “E6K increases NK cell activation protein function [14,16], whereas K53E decreases cytokine activity [16].”.
- “Few studies had used molecular dynamics (MD) simulation to analyze the structure of IL-18.” Need to cite what are those few studies
Response: We have edited into “Few studies had used molecular dynamics (MD) simulation to analyze the structure of IL-18 [16,22,23].”.
- “which was previously reported as a partial α-helix and a full α-helix” citation required.
Response: We have edited into “which was previously reported as a partial α-helix and a full α-helix (Figure 1) [14-16].”.
- Please specify the limitations of the study in the discussion section.
Response: We have added the last paragraph in the discussion section to respond this point as “However, structural analysis of the engineered IL-18 and its complex structure is required to better understand the binding affinity. Some limitations in this study need to be addressed: a) Because this in silico study is purely computational and relies on experimental structure analysis, the crystal structure of the mutant/engineered IL-18 was required for validation. b) Protein expression and purification of the engineered IL-18 IL-18 should be performed to ensure stability. c) The activity of the proposed engineered IL-18 should be tested to ensure that experimental and computational results are consistent. As a result, improving our understanding of IL structure could aid in cytokine engineering, a potentially effective treatment strategy.”.

Round 2
Reviewer 1 Report (New Reviewer)
Comments and Suggestions for Authors
The revised paper encoded ijms-3091204-R1 submitted by C. Peeyatu & colleagues has now been definitely improved in both quality/clarity of presentation and English language.
Hence, I consider this revised version to be acceptable for publication, pending only some really minor suggestions I list below:
Page2, 5th line, i.e., third yellow line: “The 6th glutamine…” can be replaced by “The E6 and K53 residues of IL-18 are crucial for IL-18R and IL-18P binding. Then, 5th yellow line: change to “The E6K mutation improves…”
Then, second yellow block, 4th and 5th lines: “the IL-18 structure influenced both structural flexibility and interaction sites [16]. (“interaction” alone is not clear with…).
Page 3, seven lines from the bottom: change to “surrounding residues”, or alternatively “surrounding amino acid (or AA) residues”.
Page 5, first line: “…loop and wild type…”, why “and”? What it means this “and”?. Then, third line: add “variants” after ‘single mutation’. Then, 4th line of paragraph 2.1.3: add a “comma” after ‘wild type’.
Page 9, six lines from the bottom: Tyr52/Thr73/Ile99 can become “Y52/T73/I99”. The other 3-letter codes can remain, being coherent with the labels in Figure 5.
Page 16, 5th line of paragraph 4.3: maybe ‘…distance “found” between…’ sounds better.
Once these changes being introduced, it is not necessary another check of mine.

Author Response
Reviewer 1
The suggestions were added as followings:
1) Page2, 5th line, i.e., third yellow line: “The 6th glutamine…” can be replaced by “The E6 and K53 residues of IL-18 are crucial for IL-18R and IL-18P binding. Then, 5th yellow line: change to “The E6K mutation improves…”
Response: We replaced the 5th line as suggested in the revised manuscript, by changing to “The E6 and K53 residues of IL-18 are crucial for IL-18R and IL-18P binding [13,14].”.We also change 5th yellow line to be “The E6K mutation improves”.
2) Then, second yellow block, 4th and 5th lines: “the IL-18 structure influenced both structural flexibility and interaction sites [16]. (“interaction” alone is not clear with…).
Response: We replaced second yellow block, 4th and 5th lines as suggested in the revised manuscript, by changing to “Prior studies found that E6K/T63A IL-18 structure influenced both structural flexibility and interaction sites [16].”.
3) Page 3, seven lines from the bottom: change to “surrounding residues”, or alternatively “surrounding amino acid (or AA) residues”.
Response: We replaced Page 3, seven lines from the bottom as suggested in the revised manuscript, by changing to “surrounding amino acid (or AA) residues”.
4) Page 5, first line: “…loop and wild type…”, why “and”? What it means this “and”?.
Response: We corrected this sentence to “The distance between the large binding site I loop in E6K/T63A increased by approximately 5 Å, compared to wild type.”.
5) Then, third line: add “variants” after ‘single mutation’.
Response: We added “variants” after ‘single mutation’ as suggested.
6) Then, 4th line of paragraph 2.1.3: add a “comma” after ‘wild type’.
Response: We added a “comma” after ‘wild type’ as suggested.
5) Page 9, six lines from the bottom: Tyr52/Thr73/Ile99 can become “Y52/T73/I99”. The other 3-letter codes can remain, being coherent with the labels in Figure 5.
Response: We corrected ‘Tyr52/Thr73/Ile99’ to “Y52/T73/I99”.
6) Page 16, 5th line of paragraph 4.3: maybe ‘…distance “found” between…’ sounds better.
Response: We changed the “distance between” to be “distances found between” as suggested.

This manuscript is a resubmission of an earlier submission. The following is a list of the peer review reports and author responses from that submission.
Round 1
Reviewer 1 Report
Comments and Suggestions for Authors
In this study, the authors analyzed the structural dynamics of IL-18 using molecular dynamics (MD) simulations. The distances from residue 63, RMSFs, etc. are shown. However, there are some major revisions:
(1) For these kinds of analyses using MD simulations, it is not appropriate to extract various properties only from a single run for each target. It should be described as an average from at least 2 or 3 runs.
(2) Intuitively, when there are large atomic fluctuations (RMSFs) in the site I region (Fig. 2), we would observe that those residues are further (or less) away from the residue 63 (Fig. 1). However, there are some mutants with large distances despite small fluctuations (E6A, E6K/T63L, E6K/T63Y etc.) and vice versa. After the above resampling, it is recommended to plot the RMSF versus distance for better understanding.
(3) Regarding hydrogen bond analyses (Figs. 3 and 4), although there are some interactions with X63, the relationships between residues (5 and 63) and the overall conformational changes are ambiguous (or unknown). It may be helpful to relate cooperative motions and local interactions by principal component analysis (PCA) (and hydrogen bond analysis).
(minor)
(4) Providing the overall structure of IL-18/IL-18Rα/IL-18Rβ shown in Figure 6A (and the sequence/alignment (Figure 1)) early in the manuscript, such as the introduction, will help readers understand this study.
(5) Where does residue number 1 start in Figure 1? (E6 is located after the 40th) It is easy to understand if this is indicated.
(6) Although Phyre2 homology modeling was used in this study, most structures are likely to be registered in the AlphaFold database if UniProt IDs are known. Why did the authors use less accurate homology modeling? Or, please verify by comparing with the AlphaFold structures.
Comments on the Quality of English LanguageThere is an expression "by about 5 Å times" on p.3. Please check scientific expressions throughout the manuscript.
Reviewer 2 Report
Comments and Suggestions for Authors
The authors present an in-silico study analyzing the effects of Thr63 alteration coupling with E6K on conformational change patterns, binding loop flexibility, and hydrogen bond networks in interleukin-18 (IL-18). I find the research topic interesting from several angles, however, the conclusions or methods do not introduce novel approaches or techniques that could significantly advance the field. Methods presented in this paper use structures derived from open-source databases/webservers and processed using a variety of tools that are straightforward but useful for computational biology. Since the simulation results are easy to follow, the results are unquestionable. Manuscripts mostly describe figures and structural confirmations rather than discussing the physical processes that take place during the formation of these structures. Although the authors explained the H-bond occupancy analysis, further validation via binding energies would help understand the binding affinity and stability of these conformations.
In conclusion, the journal IJMS seeks papers with strong significance and potential for high impact. In my opinion, the manuscript does not meet the publication criteria due to a lack of substantial new insight, lack of novelty, and weaknesses in the presentation.